# Motivation and Its Impact on Language Achievement: Sustainable Development of Ethnic Minority Students' Second Language Learning

**Shi Jiao [1,†], Jing Wang [2,\*], Xu Ma [3,†], Zheng You [4,†] and Dini Jiang [5]**

1   College of Education, Minzu University of China, Haidian District, Beijing 100081, China; jiaoshi@muc.edu.cn
2   College of Education, Zhejiang University, Zijingang, Xihu District, Hangzhou 310058, China
3   College of Economics, Minzu University of China, Haidian District, Beijing 100081, China; maxu19@muc.edu.cn
4   Graduate School of Education, Beijing Foreign Studies University, Haidian District, Beijing 100089, China; youzheng@bfsu.edu.cn
5   College of Foreign Languages and Cultures, Chengdu University, Longquanyi District, Chengdu 610106, China; jiangdini@cdu.edu.cn
\*   Correspondence: wangjing666@zju.edu.cn
†   These authors contributed equally to this work.

**Abstract:** This study examined the English learning motivation of Chinese ethnic university students. A sample of 776 undergraduates from three representative ethnic universities participated in this research. The findings indicated four types of English learning motivation: intrinsic interest, learning situation, personal development, and international communication. There were statistically significant differences between ethnic minority and Han students, and between male and female students. Moreover, the intrinsic interest motivation of Han students was significantly higher than that of ethnic minority students, and female students' overall motivation and personal development motivation were significantly higher than those of male students. Intrinsic interest motivation had a significantly positive impact on English achievement, whereas learning situation motivation had a significantly negative impact. These findings highlight the improvement of the learning situation and encouragement of intrinsic interest to enhance minority students' second language learning and sustainable development. Further research on English as a second language should consider the influence of family and ethnic background.

**Keywords:** English; motivation; achievement; ethnic minority; intrinsic interest; learning situation

## 1. Introduction

The 2030 Education Agenda aims to provide a sustainable blueprint for global education delivery to ensure equal access to education for all [1]. At the United Nations Summit on Sustainable Development, Member States formally adopted the 2030 Agenda for Sustainable Development, pledging to provide inclusive and equitable quality education for all, including females, ethnic minorities, people with disabilities, and children in vulnerable situations, emphasizing that all people should have access to lifelong learning opportunities [2].

English is a key instrument for economic growth, modernization, and globalization [3]. The motivation of English learning determines the priority of English learning. To cope with profound changes unseen in the world in a century, China today faces unprecedented opportunities and challenges. Under the sustainable development pattern, second language learning is facing profound opportunities and changes. English is not only an important medium to cope with the development of modernization and globalization [3], but also a tool to train professionals with global vision and international competence. In English

learning, English learning motivation determines students' active degree of English learning. Motivation is a central mental engine or energy center that includes effort, will, and task fulfillment [4]. Academic studies on English learning motivation mainly focus on the motivation source of English learning and the necessity of learning English [5]. For a long time, researchers have been trying to identify the components of English learning motivation, such as integration motivation and instrumental motivation, but there has not been a sufficient exploration of the inherent components of English learning motivation. A thorough study of the internal structure is helpful to explain the formation of learning motivation and its influence on English learning.

China is a multi-ethnic country with 56 ethnic groups, and nearly one-tenth of China's population are ethnic minorities. In ethnic universities, more than half of the students are from ethnic minorities [6]. Ethnic universities have an important role in serving the nation and promoting the sustainable development of ethnic minorities and ethnic regions. English is a high-status language for education and employment opportunities. Equal opportunities for second language learning will help the country achieve equal education for students from different ethnic minorities [7]. English learning of minority college students plays an important role in the sustainable development of minority and educational equity. The quality of English teaching has always been the focus and challenge of ethnic education and research. The learning environment and cultural background of ethnic minority college students are very different from those of non-ethnic group (Han) students [8]. The learning and living environment of minority college students is far away from their ethnic-cultural background, their English learning experience is more complicated and difficult than that of Han students, and their English learning motivation has certain particularities compared with Han students. For example, ethnic minorities, such as Tibetans and Mongolians, usually speak their mother tongue as their first language, Mandarin as their second, and English as their third. Compared with non-ethnic-group students, minority students have more difficulties in second language learning [7].

However, little is known about the English learning motivation of minority college students. Ethnic minorities are an important part of the country, even though they represent a smaller portion of the population. Research on ethnic minorities seems to be as important as that on non-ethnic groups. Universities play a key role in setting sustainable development goals and promoting sustainable development in education [9]. Ethnic universities contain more than half minority college students. Therefore, study of the English learning motivation of ethnic college students is an important reference for study of minorities' second language learning and the sustainable development of education. Previous studies indicate that minority students are more likely to be disadvantaged in their studies. "As an ethnic minority, you just have to work twice as hard". It indicated that ethnic minorities have to work harder if they wanted to stand out [10]. Ethnic minorities have less chance of completing higher education than the majority [11]. Study on minority students is very meaningful to the equality and sustainability of not only education, but the whole society as well. Therefore, a study on the English learning motivation of ethnic undergraduates seems particularly important.

## 2. Literature Review

Motivation is a kind of central mental engine or energy center that encompasses effort, will, and task enjoyment [4]. English learning motivation is an important index to measure students' English learning. It is defined by Dörnyei and Ottó [12] as the cumulative arousal of a person's dynamic changes that initiates, guides, coordinates, amplifies, terminates, and evaluates the cognitive and motor processes to select, prioritize, achieve, and execute the original wishes and desires. English learning motivation is both a spiritual "engine" and an emotional attitude. Therefore, English learning motivation is an important reference that cannot be ignored in second language learning. Previous studies on motivation in second language learning can mainly be divided into three aspects: motivation composition, ethnic and gender differences, and the motivational effect on English achievement.

### 2.1. Motivation Composition

Early research on language learning motivation follows Gardner's (1985) social educational model, which divides second language learning motivation into integrativeness and instrumentality [13,14]. Some scholars also take internal motivation and external motivation as a paradigm to discuss language learning based on self-determination theory [15–17]. However, with the development of globalization, English has broken away from the limitations of traditional regions. It seems difficult to solve the fundamental problem of English learning motivation only from the perspective of integration and instrumentality, or even intrinsic and extrinsic motivation [18]. Dörnyei and Ushioda [19] redefined language learning motivation from the perspective of self-identity and proposed the "L2 motivational self system", which includes three aspects: the ideal L2 self, the ought-to L2 self, and the L2 learning experience [20].

Gardner and Lambert [21] suggested that instrumental and integrative motivation were the main motivations for second language learning. Instrumental motivation is defined as the utilitarian acquisition of second language proficiency, for example, to get a higher salary or a better job. Integrative motivation is defined as the willingness to be like valued members of the language community [22]. Since this orientation was proposed, instrumentality and integrativeness have been the focus of research on second language learning motivation. In recent years, studies on second language learning motivation have shifted their focus from instrumental and integrative discussions to more complex socio-cultural situations and interactions between learners. The influence of learning situations and personal vision on second language learning motivation has become more and more important.

Csizér and Dörnyei [23] stated that integrativeness is the most important motivation for second language learning, mediating the influence of all the other factors. A recent study [24] asserted that most second language learners possessed integrative learning motivation, and people with high integrative motivation could easily achieve high results. However, another report indicated that instrumental learning motivation was the preference for second language learners [25]. This argument was supported by Long et al. [26], who examined the main causes of change in students' English learning motivation. Dos Santos [27] advocated that both integrative motivation and instrumental motivation could promote learning. This idea echoed the finding of Gardner and MacIntyre [28] that motivation has dual components, integrativeness and instrumentality, both of which can improve language learning.

Integrative motivation and instrumental motivation are not mutually exclusive; generally, second language learning involves a composition of various levels. Therefore, it is hard to classify language learning motivation as a precise type, the importance of which relies on the situation of the study [21]. At the same time, intrinsic motivation and extrinsic motivation have become another perspective in language learning research [29,30]. It is undeniable that the research on the internal and external division of motivation has important implications for individuals' willpower and academic achievement [31,32]. However, this division makes it difficult for us to understand the learning experience of second language learners, such as the role of learning situations, which is particularly important for minority students [33]. In fact, both intrinsic and extrinsic motivation could find their explanations from the three levels of the "L2 motivational self system" [19]. Therefore, using the theory of the "L2 motivational self system" could help us answer the unsolved mysteries of language learning motivation.

In the past decade, the "L2 motivational self system" has been widely applied in the global environment and has become the mainstream paradigm for the study of language motivation [29]. You and Dörnyei [34] conducted a large-scale survey on Chinese language learners from the perspective of the "L2 motivational self system", and found that the motivation types of Chinese second language learners are instrumentality-promotion, cultural interest, travelling, instrumentality-prevention, parental expectation, and expected efforts. Moskovsky et al. [35] explored the relationship between Dörnyei's (2009) L2 motivational

self system and the second language proficiency of Saudi L2 learners. The study classified second language learning motivation according to the level of the ideal L2 self, the ought-to L2 self, and the L2 learning experience [19,20], indicating that the components of the L2 motivational self system are good predictors of learners' intention to learn. Gao, Zhao, Cheng, and Zhou [3] found seven main types of English learning motivation for university students in China: intrinsic interest, immediate achievement, going abroad, learning context, social responsibility, personal development, and information media, which were in line with the aspects of the "L2 motivational self system". Based on the "L2 motivational self system", another study investigated and analyzed the motivational antecedents of Nepalese learners' English learning motivation [36].

### 2.2. Influence of Ethnicity and Gender

Guo and Leung [37] compared the learning motivation between Chinese Miao and Han students, indicating motivational difference by ethnicity. A recent study [38] examined the motivational differences among different ethnic groups, indicating that there was a positive correlation between intrinsic motivation and English academic self-efficacy. Parental advising had positive and significant effects on intrinsic motivation toward English learning and engagement for Caucasian and African American students, whereas it had no positive effects on Asian and Hispanic students. D'Lima et al. [39] argued that African-American and Caucasian students had greater self-efficacy in academic achievement than Asian-American university students. African American and Asian American students initially had more extrinsic motivation than Caucasian students. Mastery orientation and intrinsic and extrinsic motivation were positively correlated with academic performance.

A previous study indicated that female students' learning motivation was higher than that of male students. Yang and Quadir [40] supported this statement in a recent study. Oga-Baldwin and Nakata [41] argued that female students made more efforts than male students and therefore performed better in academic achievements. Female students had more extrinsic motivation and mastery orientation than male students, who are more focused on performance [39]. You and Dörnyei [34] asserted that female students had stronger visions for personal development than boys, and female students studied harder than male students. As a result, girl students were more inclined to obtain higher achievement than boys.

### 2.3. Motivational Effect on English Achievement

It is generally believed that academic achievement cannot be separated from the influence of learning motivation [42]. The higher the motivation, the better the academic performance [5]. Conversely, the weaker the motivation, the weaker the academic performance [43]. Obviously, motivation had a significant predictive effect on academic achievement. Previous studies found that intrinsic interest motivation had a significant predictive effect on students' English learning performance [44,45]. This suggested that intrinsic interest played a positive role in improving academic achievement [17]. At the same time, some studies have argued that ethnic minority students' motivation in a learning situation had a significant negative impact on academic achievement [39]. This implies that learning situations tend to put minority students at a disadvantage. Therefore, situational motivation is an important aspect of ethnic minority student motivation research. Wang and Rao [46] indicated that Chinese students' learning motivation is influenced by social responsibility, and responsibility motivation had a predictive effect on students' grades. Canning et al. [47] advocated that teachers' mindset beliefs predicted students' motivation and academic achievement.

Although there are debates about the research on second language learning motivation, the overall trend is shifting from simple individual psychological factors to complex socio-cultural situations and interactions between individuals. Cheng et al. [48] found that individual and social situation factors could change the motivation, test anxiety, and test performance of test-takers. Motivation and test anxiety interacted with individual and

social context factors. In different test contexts, the importance and purpose of the test affected the motivation and test anxiety of the subjects. Previous studies have advocated that the surrounding environment plays an important role in language learning motivation [27,49]. Busse and Walter [50] examined first-year university students in the UK who learned German, and the data showed that there was a strong relationship between second language learning and the learning situation, especially with second language courses and teachers. Hennebry-Leung and Xiao [51] indicated that teachers play an important role in second language learning motivation. Courtney [52] investigated the motivation of language learning and the development of second language level, suggesting that a sudden change of teaching method might have a negative impact on the attitude and motivation of learners in junior high school.

Based on the theoretical framework of the L2 motivational self system, You and Dörnyei [34] divided motivation into three levels: the ideal L2 self, the ought-to L2 self, and the L2 learning experience. Dörnyei [53] indicated that the L2 learning experience was a kind of situational motivation, and it was a powerful predictor of students' motivational behavior. The L2 learning experience was considered as the perceived quality of language learners' engagement in all parts of the language learning process. For ethnic minorities, situational motivation had the highest influence on their study [10]. Motivation is closely related to the learning situation, and the classroom situation has a great impact on students' motivation and performance to learn second languages [54]. Therefore, when examining ethnic minorities, it is particularly important to study situational motivation and its influence on ethnic minorities' academic performance.

On the whole, the research on English learning motivation in China has also begun to shift from the debate on composition to a discussion on motivational effects and academic performance. In addition, learning situation, social background, and learning involvement are becoming important factors affecting second language learning. The previous studies on second language learning motivation mainly focus on the dominant ethnic groups (Han), and few studies have paid enough attention to minority students, especially university students of ethnic minorities. The present research on students' second language learning motivation has solved the problem of motivational types. However, the internal structure of minority students' motivation has not attracted enough attention. Similarly, ethnic differences, gender differences, and the motivational effects on language performance have not been fully explored.

The previous model of motivation composition is usually called motivation type, which is a classification based on exploratory factor analysis (EFA) [55,56]. EFA aims to confirm the factor structure of the scale, which is biased towards theoretical output but lacks a test for the theoretical framework [57]. Confirmatory factor analysis (CFA) is usually used to verify the appropriateness of the measurement model. CFA is a basic component of structural equation modeling (SEM) [58]. CFA analysis is often tested by structural equation modeling, which can test the authenticity and appropriateness of construct validity [59]. On the basis of EFA, CFA is added to the present study in order to obtain the best factor structure of the questionnaire and establish the construct validity of the questionnaire. CFA can be used to test the structural validity of motivation, so as to determine the classification structure of motivation more accurately [60]. Therefore, the combination of CFA and EFA goes beyond the exploration of motivation types [61]. Here, we call it the internal structure of motivation. In this article, we seek to analyze the internal structure of ethnic minority undergraduates' learning motivation, observing the differences between dominant ethnic groups (Han) and ethnic minorities, and male students and female students. Beyond that, we try to discover the motivational impact on English achievement.

### 2.4. Purpose of the Study

The purpose of this study is to explore the motivation of ethnic minority university students in English learning, focusing on the internal structure, ethnic and gender differences, and motivational impact on English achievement. Hence, we can deepen the

understanding of the English learning motivation of Han and ethnic minority students and provide some references for second language teaching and the sustainable development of ethnic minority college students.

The learning and living environment of ethnic undergraduates are far from their ethnic-cultural background, so their English learning motivations have certain particularities compared to non-ethnic undergraduates. This study selects Chinese ethnic undergraduates as the research participants and aims to investigate the English learning motivation of ethnic minority students. At the same time, through comparison, this paper explores the particularities of the English learning motivation of ethnic undergraduates, examining the motivational effects on language performance. Accordingly, the present study focused on the following research questions:

- What are the internal structures of English learning motivation for undergraduates in Chinese ethnic universities?
- Are there any differences between dominant ethnic groups (Han) and ethnic minority students, or male students and female students?
- What is the motivational impact on minority students' English academic performance?

## 3. Theoretical Frameworks

In this study, we adopted the L2 motivational self system proposed by Dörnyei and Ushioda [19]. The L2 motivational self system was derived from the possible selves theory in the L2 field and social psychology, and includes three aspects: the ideal L2 self, the ought-to L2 self, and the L2 learning experience [19,20]. The ideal L2 self refers to learners' strong self-image of learning a second language in an ideal environment. It reflects an ideal state that second language learners can achieve in future learning and is a powerful motive for second language learning. The traditional integrative and instrumental motives belong to this aspect. The ought-to L2 self reflects the belief in oneself, and it avoids negative consequences by fulfilling the expectations of others, mainly embodied in instrumental prevention and parental expectations. This part reflects the extrinsic expectation of learning rather than its intrinsic vision. Therefore, this component is mainly reflected in the extrinsic tool motivation. The L2 learning experience is a kind of situational and executive motivation, which is related to the immediate learning environment and experience. It includes teachers, peer groups, curriculums, and successful experiences [34]. Meanwhile, the L2 learning experience is the perceived quality of second language learners' participation in the language learning process and is the strongest predictor of motivated behavior [53].

The L2 motivational self has been widely used to explain learners' expected learning effort and motivational behavior over the past two decades. For instance, through a large-scale hierarchical questionnaire survey, You and Dörnyei [34] outlined the motivational tendencies of Chinese English learners. You et al. [62] investigated the vision and intention of motivation of English learners and found that a change of vision and intention affects the development of motivation. Tseng et al. [63] developed a tool based on a four-dimensional model that uses learners' self-image as a means of exploring language motivation, which plays a key role in sustaining second language learners' efforts.

Motivational studies have begun to shift from a focus on the ideal L2 self level and ought-to L2 self level to the L2 learning experience, especially the most important part, the learning situation [34]. For ethnic minorities, situational components have been proved to be influential and crucial to second language learning. Therefore, the L2 learning experience component regarding learning situations needs to be further explored. Based on the L2 motivational self system [19], this study aims to explore the internal structure of ethnic undergraduates' English learning motivation, examining the particular situation of ethnic minority students and investigating the motivational effect on the second language achievement.

## 4. Materials and Methods

### 4.1. Participants

In this study, we took random samples by selecting three representative ethnic universities, which have a high proportion of ethnic minority students. These three universities are located in the east, south, and northwest part of China, respectively. The participants are both Han and ethnic minority students. According to the ratio of 1:100, samples of undergraduates from universities in humanities, social sciences, and natural sciences were selected, covering 11 disciplines. This study was anonymous, and participation was voluntary. A total of 800 questionnaires were distributed, and 776 of them with valid responses were returned. The response rate was 97%. The sample details are shown in Table 1.

**Table 1.** Survey sample details (n = 776).

| Variable | | Frequency | Percentage |
|---|---|---|---|
| Gender | Male | 260 | 33.5% |
| | Female | 516 | 66.5% |
| Ethnicity | Ethnic minorities | 457 | 58.9% |
| | Han | 319 | 41.1% |
| University location | East | 166 | 21.4% |
| | South | 244 | 31.4% |
| | Northwest | 366 | 47.2% |
| Disciplines | Humanities | 156 | 20.1% |
| | Social sciences | 379 | 48.8% |
| | Natural sciences | 241 | 31.1% |

The participants of ethnic minorities selected in the study come from different ethnic groups, including Chinese Mongolian, Tibetan, and other ethnic groups, mainly representing the macro group of ethnic minorities. This study does not compare each ethnic group separately. Ethnic minority college students often come from frontier ethnic areas, usually located in remote and economically underdeveloped areas, which are far away from the developed eastern coastal areas [64]. It is generally believed that the majority of students have obvious advantages over minority students in second language learning [65]. The research on ethnic education takes all ethnic minorities as a sample to compare with the mainstream ethnic Han [6]. Therefore, we can take minority college students as a research subject in this study. The minority students are the combination of 55 ethnic groups compared to Han college students.

### 4.2. Instruments

According to the Questionnaire on the English Learning Motivation Types for Chinese Students from Gao, Zhao, Cheng, and Zhou [3], the initial project of The Questionnaire on English Learning Motivation of Ethnic College Students is adapted and carried out. Some duplicate items were combined, and those that did not conform to this study were deleted. Overall, 30 items were determined to constitute the initial questionnaire. A five-point Likert scale (from 1 = very inconsistent to 5 = very consistent) was used to evaluate English learning motivation. The higher the score, the stronger the English learning motivation.

Moreover, focusing on ethnic undergraduates, appropriate adjustments were made for some of the topics. Prior to the distribution of the questionnaires, 90 students were selected for a pilot study using the initial questionnaire. Irrelevant items were deleted, and the revised questionnaires were administered.

Finally, questionnaires were administered among three sampled ethnic universities. These three representative ethnic universities came from three different areas, representing samples of well-developed and less-developed regions of China.

*4.3. Data Analytic Procedure*

Structural equation modeling (SEM) was conducted for statistical analysis of the data. SPSS 20.0 software (IBM SPSS Statistics for Windows, Version 20.0., IBM Corp., Armonk, NY, USA, 2011.) was used to conduct statistical analysis of the data. On this basis, exploratory factor analysis (EFA) was carried out on the English learning motivation of ethnic undergraduates. Amos 20.0 software (Arbuckle JL. Amos, Version 20.0. [Computer Program], IBM SPSS, Chicago, IL, USA, 2011.) was used to conduct confirmatory factory analysis (CFA) to examine the model fit.

## 5. Results

*5.1. Item Analysis*

Half of the data of the test (n = 388) was randomly selected from the total sample (n = 776) for project analysis and exploratory factor analysis. Through correlation test, the correlation between each topic and the total score was calculated, and the items with insignificant correlation coefficient were deleted, together with the items with correlation coefficient less than 0.3.

According to the item analysis of the significance test of criteria ratio (CR), the highest 27% of the total score of the questionnaire and the lowest 27% of the total score were taken as the critical values of high and low scores, and the questions with insignificant correlation coefficients were deleted. In this study, a total of three items were deleted (factors with factor load lower than 0.50 were deleted), which were items 1, 2, and 12, leaving 27 items.

*5.2. Exploratory Factor Analysis of the Internal Structure*

Exploratory factor analysis (EFA) was carried out on the questionnaire data by SPSS20.0 statistical software, and spherical test was conducted on KMO and Bartlett, and it was found that KMO = 0.879 and Bartlett spherical test results were significant (approximately CMIN = 4220.354, *df* = 253, *p* = 0.000), indicating that there were common factors among the correlation matrices of the data, and the questionnaire was suitable for exploratory factor analysis. The main component analysis method was used to extract factors and common factors from the 27 questions of the questionnaire, and the initial factor load matrix was obtained. Combining data statistics and theoretical analysis results, the four items of 3, 4, 6, and 11 were deleted, leaving 23 items. Exploratory factor analysis was conducted on the remaining data again, and four factors were extracted according to the standard with eigenvalue greater than 1. The total variance of the cumulative interpretation was 58.992% (See Table 2).

As can be seen from Table 2, the English learning motivation of ethnic undergraduates included four dimensions: intrinsic interest, learning situation, personal development, and international communication. Intrinsic interest refers to "interest in English-speaking countries, interest in language learning, love of language itself, love of English literature, love of English songs/movies, let the world know our country/nation, understand the world's economic/technological/educational development, and do our best for the prosperity of the country/nation". Learning situation includes "English classes, English teachers, and English textbooks". Personal development includes "communication tools, stepping stone to one's success in life, achieving a sense of accomplishment, getting a good job, education/accomplishment symbols, and studying other majors/subjects". International communication includes "going abroad to study/work, going abroad to experience culture, emigrating to a foreign country, living up to parents/family expectations".

**Table 2.** Exploratory factor analysis (EFA) results.

| The Questionnaire Items | 1 Intrinsic Interest | 2 Learning Situation | 3 Personal Development | 4 International Communication | Communality |
|---|---|---|---|---|---|
| 18. Interested in English-speaking countries. | 0.843 | | | | 0.730 |
| 19. A love of language learning. | 0.820 | | | | 0.698 |
| 21. Love the language. | 0.810 | | | | 0.660 |
| 23. Love English literature. | 0.807 | | | | 0.666 |
| 20. Like English songs/movies. | 0.785 | | | | 0.622 |
| 22. Let the world know about our country/nation. | 0.772 | | | | 0.616 |
| 17. Understand the world's economic/technological/educational developments. | 0.612 | | | | 0.454 |
| 24. Do our best for the prosperity of the country and the nation. | 0.593 | | | | 0.499 |
| 10. Like fellow students in the English class after entering university. | | 0.798 | | | 0.667 |
| 7. Like English teacher after entering university. | | 0.790 | | | 0.638 |
| 9. Like English textbooks after entering university. | | 0.767 | | | 0.624 |
| 5. Like English teacher before entering university. | | 0.752 | | | 0.573 |
| 8. Like English class after entering university. | | 0.736 | | | 0.565 |
| 13. Communication tools. | | | 0.792 | | 0.643 |
| 29. Stepping stone to one's success in life. | | | 0.754 | | 0.596 |
| 14. Get a sense of accomplishment. | | | 0.728 | | 0.591 |
| 16. Get a good job. | | | 0.671 | | 0.457 |
| 30. Symbol of education/culture. | | | 0.648 | | 0.453 |
| 15. Study other majors/subjects. | | | 0.635 | | 0.482 |
| 26. Study/work abroad. | | | | 0.804 | 0.680 |
| 27. Go abroad to experience culture. | | | | 0.763 | 0.657 |
| 28. Emigrate to a foreign country. | | | | 0.680 | 0.550 |
| 25. Live up to your parents/family expectations. | | | | 0.570 | 0.447 |
| Eigenvalue | 7.115 | 3.023 | 1.989 | 1.441 | |
| Contribution rate % | 30.934 | 13.143 | 8.647 | 6.267 | |
| Cumulative contribution rate % | 30.934 | 44.077 | 52.724 | 58.992 | |

The table reports the results of oblique rotation. Oblique rotation allows the correlation between factors, and the factors obtained by orthogonal rotation are independent of each other.

### 5.3. Confirmatory Factor Analysis

According to the exploratory factor analysis results, Amos 20.0 statistical software was used to carry out confirmatory factor analysis (CFA) on the other half of the data that was formally tested (n = 388). To test the four-factor model fitting of university students' English learning motivation, this was composed of intrinsic interest, learning situation, personal development, and international communication. The results of confirmatory factor analysis showed that in the fitting indexes of the structural equation modeling (SEM), CMIN/$df$ = 2.855, RMSEA < 0.08, SRMR < 0.08, and CFI, IFI, and TLI were all greater than 0.90, indicating that the CFA model had a good structural fit [66] and verifying the multidimensional structural hypothesis of the questionnaire on English learning motivation of ethnic undergraduates, as shown in Table 3.

**Table 3.** Confirmatory factor analysis (CFA) fitting index of the model.

| Fitting Index | CMIN | *df* | CMIN/*df* | RMSEA | SRMR | CFI | IFI | TLI |
|---|---|---|---|---|---|---|---|---|
| Value | 608.022 | 213 | 2.855 | 0.069 | 0.066 | 0.922 | 0.922 | 0.907 |
| Standard | | | <5.0 | <0.08 | <0.08 | >0.90 | >0.90 | >0.90 |

### 5.4. Reliability Analysis

After screening, the internal consistency coefficient (Cronbach's $\alpha$) of the total questionnaire was 0.910, and the reliability of the four dimensions of intrinsic interest, learning situation, personal development, and international communication were 0.910, 0.843, 0.844, and 0.738, respectively. According to Gao, Zhao, Cheng, and Zhou [3], the overall internal consistency coefficient (Cronbach's $\alpha$) of the original questionnaire on university students' English learning motivation was 0.84, which was 0.07 higher than that of Gao's. This study showed that the questionnaire on English learning motivation of ethnic undergraduates had a good reliability index.

Confirmatory factor analysis (CFA) showed that the standard regression weights (SRW) of the underlying variable corresponding to each item ranged from 0.47 to 0.92 and were higher than 0.45 ($p < 0.001$). The construct reliability (CR) of intrinsic interest, learning situation, personal development, and international communication were all significantly higher than 0.70. From this, it could be seen that CFA statistical results supported the statistical results of EFA, and the convergent validity of the questionnaire on English learning motivation of ethnic undergraduates was good, as shown in Table 4.

**Table 4.** Confirmatory factor analysis (CFA) results.

| Latent Variables | Observation Variable | EFA's Factor Loading | Cronbach's Alpha | The CFA's SRW | AVE | CR |
|---|---|---|---|---|---|---|
| Intrinsic interest | 18. Interested in English-speaking countries. | 0.843 | | 0.83 | | |
| | 19. A love of language learning. | 0.820 | | 0.71 | | |
| | 21. Love the language. | 0.810 | | 0.75 | | |
| | 23. Love English literature. | 0.807 | | 0.78 | | |
| | 20. Like English songs/movies. | 0.785 | | 0.85 | | |
| | 22. Let the world know about our country/nation. | 0.772 | 0.910 | 0.83 | 0.5836 | 0.9178 |
| | 17. Understand the world's economic/technological/educational developments. | 0.612 | | 0.76 | | |
| | 24. Do our best for the prosperity of the country and the nation. | 0.593 | | 0.69 | | |
| Learning situation | 10. Like fellow students in the English class after entering university. | 0.798 | | 0.65 | | |
| | 7. Like English teacher after entering university. | 0.790 | | 0.92 | | |
| | 9. Like English textbooks after entering university. | 0.767 | 0.843 | 0.66 | 0.5353 | 0.8494 |
| | 5. Like English teacher before entering university. | 0.752 | | 0.75 | | |
| | 8. Like English class after entering university. | 0.736 | | 0.64 | | |
| Personal development | 13. Communication tools. | 0. 792 | | 0.79 | | |
| | 29. Stepping stone to one's success in life. | 0.754 | | 0.66 | | |
| | 14. Get a sense of accomplishment. | 0.728 | | 0.80 | | |
| | 16. Get a good job. | 0.671 | 0.844 | 0.69 | 0.5095 | 0.8605 |
| | 30. Symbol of education/culture. | 0.648 | | 0.59 | | |
| | 15. Study other majors/subjects. | 0.635 | | 0.73 | | |
| International communication | 26. Study/work abroad. | 0.804 | | 0.76 | | |
| | 27. Go abroad to experience culture. | 0.763 | | 0.91 | | |
| | 28. Emigrate to a foreign country. | 0.680 | 0.738 | 0.47 | 0.4743 | 0.7709 |
| | 25. Live up to your parents/family expectations. | 0.570 | | 0.52 | | |

*5.5. Exploration of Internal Relationships among the Factors of English Learning Motivation*

The results of the correlation coefficient matrix between each factor of English learning motivation and the total score are shown in Table 5. All the factors were correlated with each other, and the factors were highly correlated with the total score; the correlation coefficient was between 0.547 and 0.884, which was higher than the correlation among the factors, which also verified the good structural validity of the scale.

**Table 5.** Correlation matrix among factors and between factors and total score.

| Factors | Intrinsic Interest | Learning Situation | Personal Development | International Communication | Total Score |
|---|---|---|---|---|---|
| Intrinsic interest | 1 | | | | |
| Learning situation | 0.264 ** | 1 | | | |
| Personal development | 0.615 ** | 0.185 ** | 1 | | |
| International communication | 0.564 ** | 0.241 ** | 0.459 ** | 1 | |
| Total score | 0.884 ** | 0.547 ** | 0.770 ** | 0.720 ** | 1 |

Note: ** means $p < 0.01$.

*5.6. Characteristics of Students' English Learning Motivation*

5.6.1. English Learning Motivation in Each Dimension

The mean results of the total scale and each factor of English learning motivation are shown in Table 6. Among them, the mean value of personal development motivation was the highest ($M = 3.79$), and the mean value of learning situation motivation was the lowest ($M = 2.93$), which indicated that university students of ethnic universities had the strongest personal development motivation and the weakest learning situation motivation.

**Table 6.** Mean values of each dimension and total score.

| Factors | *n* | *M* | *SD* |
|---|---|---|---|
| Intrinsic interest | 776 | 3.33 | 0.85 |
| Learning situation | 776 | 2.93 | 0.87 |
| Personal development | 776 | 3.79 | 0.73 |
| International communication | 776 | 3.02 | 0.82 |
| Total score | 776 | 3.31 | 0.61 |

5.6.2. Ethnic Differences in Motivation

As shown in Table 7, through the independent sample *t*-test, it was concluded that there were significant differences in the intrinsic interest motivation between minority students and Han students in ethnic universities, but no significant differences in other motivations were found. As can be seen from Table 7, the intrinsic interest motivation of Han students ($M = 3.41$) was significantly higher than that of minority students ($M = 3.28$). Combined with the analysis of independent sample *t*-test, it could be verified that the intrinsic interest motivation of Han students was significantly higher than that of ethnic minority students ($p = 0.030$).

**Table 7.** Ethnic differences in intrinsic motivation of interest: independent sample *t*-test results.

| Factors | Ethnicity | *n* | *M* | *SD* | *F* | *t* | *p* |
|---|---|---|---|---|---|---|---|
| Intrinsic interest | Ethnic minorities | 457 | 3.28 | 0.88 | 0.261 | 2.172 | 0.030 |
| | Han | 319 | 3.41 | 0.81 | | | |

5.6.3. Gender Differences in Motivation

As can be seen from Table 8, through the independent sample *t*-test, it could be concluded that students of different genders had significant differences in the total value of English learning motivation and personal development motivation, but no significant

differences in other motivations were found. The total motivational value of female students (*M* = 3.36) was significantly higher than that of male students (*M* = 3.21). The personal development motivation of female students (*M* = 3.88) was significantly higher than that of male students (*M* = 3.59). Combined with the analysis of the independent sample *t*-test, we found that the total motivation of English learning (*p* = 0.002) and personal development motivation (*p* = 0.000) of female students was significantly higher than that of male students.

**Table 8.** Gender differences: independent sample *t*-test results.

| Factors | Gender | *n* | *M* | *SD* | *F* | *t* | *p* |
|---|---|---|---|---|---|---|---|
| Personal development | Male | 260 | 3.59 | 0.81 | 6.613 | 5.034 | 0.000 |
| | Female | 516 | 3.88 | 0.67 | | | |
| Total scale | Male | 260 | 3.21 | 0.68 | 2.911 | 3.143 | 0.002 |
| | Female | 516 | 3.36 | 0.57 | | | |

*5.7. Motivational Influence on English Achievement*

The test results show the effects of English learning motivation on English achievement of ethic university students, and the results of multiple linear regression are shown in Table 9.

**Table 9.** Results of multiple linear regression.

| Model | Non-Standardized Coefficient | | The Standard Coefficient | *t* | Sig. |
|---|---|---|---|---|---|
| | **B** | **Standard Error** | | | |
| (Constant) | 1.155 | 0.147 | | 7.876 | 0.000 |
| Intrinsic interest ($X_1$) | 0.119 | 0.041 | 0.145 | 2.931 | 0.003 |
| Learning situations ($X_2$) | −0.063 | 0.030 | −0.079 | −2.121 | 0.034 |
| Personal development ($X_3$) | 0.039 | 0.044 | 0.040 | 0.885 | 0.376 |
| International Communication ($X_4$) | −0.005 | 0.038 | −0.006 | −0.144 | 0.886 |
| $R^2$ | $R^2_{Adj}$ | *F* | | | |
| 0.028 | 0.023 | 5.547 | | | |

A multiple linear regression model was conducted for statistical analysis in this study, and the equation expression of the model was as follows:

$$Y = 1.155 + 0.119X_1 - 0.063X_2$$

$$R^2_{Adj} = 0.023, \ F = 5.547 \ (p = 0.000)$$

The multiple linear regression indicated that intrinsic interest and learning situation motivation had a significant impact on the ethnic minority students' English achievement. Intrinsic interest motivation had a positive effect on English performance, whereas learning situation motivation had a negative impact on it. For each unit change in $X_1$, $Y$ changes 0.119; for each unit change in $X_2$, $Y$ changes −0.063 (negative impact).

## 6. Discussion

*Research Findings*

This study examined the motivational structure of ethnic undergraduates' English learning, the ethnicity differences, the gender differences, and the motivational impact on English achievement.

A major contribution was the internal structure of the English learning motivation of ethnic undergraduates. The findings indicated that the English learning motivation of ethnic undergraduates mainly included four types of motivation: intrinsic interest, learning situation, personal development, and international communication. Intrinsic interest

motivation referred to students' interest in the English language, culture, science, and technology. The motivation of the learning situation was mainly the interest in English courses, teaching materials, and teaching methods. Personal development motivation referred to the instrumental purpose of students' learning. Good English language competency could facilitate students' opportunities in job hunting and career development. The motivation for international exchange referred to students' planning for study abroad programs, as well as work and living experience in foreign countries.

There were significant differences in participants' English learning motivation according to their ethnicity and gender. The intrinsic interest motivation of Han undergraduates was significantly higher than that of minority undergraduates. Through the independent sample *t*-test, it was found that the intrinsic interest motivation of Han students was significantly higher than that of minority students. At the same time, female students were significantly higher than male students in terms of the total score of English learning motivation and personal development motivation.

According to the multiple linear regression, this study indicated that the English learning motivation of ethnic minority undergraduates had a significant impact on English achievement. Intrinsic interest motivation had a significantly positive effect, while learning situational motivation had a negative impact.

*Research Question 1: The Internal Structure of English Learning Motivation of Ethnic Undergraduates*
*Four motivation components were found regarding ethnic undergraduates.*

According to the questionnaire structure of Gao, Zhao, Cheng, and Zhou [3], the study firstly constructed the initial items of the questionnaire on the English learning motivation of ethnic undergraduates. Secondly, four dimensions of motivation were constructed through exploratory factor analysis (EFA). In addition, combined with confirmatory factor analysis (CFA), this paper tested the structural validity, and finally determined the four structures of ethnic undergraduates' English learning motivation.

The results of this study were in line with the findings of Gao, Zhao, Cheng, and Zhou's [3] research on the types of English learning motivation for university students, indicating intrinsic interest, learning situation, personal development, and international communication for ethnic undergraduates. However, the factors of achievement, social responsibility, and information medium were not prominent in English learning motivation, which was the difference between students of ethnic and non-ethnic universities. Admittedly, Gao, Zhao, Cheng, and Zhou's [3] research included non-ethnic and ethnic universities; however, it did not discuss ethnic universities and ethnic minority students. Therefore, this study made a significant contribution to the research on ethnic universities and ethnic minority students.

Intrinsic interest motivation reflected the interest aspect of the ideal L2 self level of Dörnyei and Ushioda [19]. The motivation of intrinsic interest indicated ethnic undergraduates' appreciation of the second language itself. This echoed the findings of Brown [67], You and Dörnyei [34], and Gao, Zhao, Cheng, and Zhou [3]. The motivation of intrinsic interest echoed the integrative motivation of Gardner [68] as well. Learners with intrinsic interest motivation were inclined to hold a positive ideal of self-image, maintaining the same positive attitude towards learning a second language. Therefore, for ethnic minorities, intrinsic interest played a noticeable and sustainable role in second language learning.

Learning situation motivation represented the L2 learning experience according to the L2 motivational self system from Dörnyei and Ushioda [19]. Learning situation referred to the course-related factors of ethnic undergraduates' learning, which mainly reflected the relevant environmental supports of teaching, such as teachers and courses [69]. Learning situation was aligned with the "learning situation level" of Waninge, Dörnyei, and De Bot [54] and the "learning situation motivation" of Gao, Zhao, Cheng, and Zhou [3]. A recent study suggested that situational motivation had the strongest impact on ethnic minority stu-

dents. This study confirmed previous research results showing that learning situation in L2 learning experience constituted the most powerful predictive tool of motivational behavior.

Personal development motivation represented the ideal L2 self level based on the L2 motivational self system [19]. The personal development motivation of ethnic undergraduates in English learning embodied a more practical tool. This was consistent with the "personal development motivation" of Gao, Zhao, Cheng, and Zhou [3]. Consistent with the result of Dos Santos [27], ethnic minority college students had a strong motivation to seek personal promotion and sustainable development. It also verified instrumental motivation according to social psychological theory [68]. Individuals who had a high need for achievement paid great attention to excellence itself, leading to pioneering achievements, such as language learning, working on these tasks with higher intensity and persisting in the face of failure [70]. On the contrary, ethnic university undergraduates had no obvious motivation for test-taking. The ethnic minority college students had been far away from the college entrance examination (gaokao) for a long time and had no significant examination pressure. Therefore, the examination motivation is not obvious.

The results confirmed both the ideal L2 self and ought-to L2 self components from the theory of Dörnyei and Ushioda [19]. Ethnic undergraduates held obvious international communication motivation, but also had to meet the expectations of parents. The international communication here was also in line with Gardner's [68] integrative motivation. The international exchange of ethnic undergraduates was mainly through overseas programs and exchange experiences, which corresponded to the "motivation to go abroad" of Gao, Zhao, Cheng, and Zhou [3]. Here, we defined it as "motivation for international communication" rather than "motivation for going abroad", which was more in line with the characteristics and sustainable development of the new world. With the advancement of China's reform and opening up, as well as the development of globalization, the international exchange activities of ethnic minority students have become increasingly frequent. Therefore, the international communication motivation proved to be important not only for ethnic university students, but for the sustainability of universities as well.

*Research Question 2: Differences in English Learning Motivation among Ethnic Undergraduates*
*The intrinsic interest motivation of Han students was obviously higher than that of minority students.*

Previous studies have indicated that intrinsic interest has a significant predictive effect on academic performance [17,24]. However, the present intrinsic interest of minority university students was obviously lower than that of Han students, which would put the academic performance of minority undergraduates in a disadvantageous position. This study examined the difference between Han and minority undergraduates. Lamb [71] believed that social background and economic background affected learners' English learning motivation. Family background had a great influence on minority university students, which made Han undergraduates more dominant in learning English than minority university students, so they had a higher sense of achievement, resulting in significant differences in intrinsic interest motivation. At the same time, Han undergraduates were more likely to stay in more developed eastern regions, where the demand for English was higher than that of ethnic minorities. Therefore, Han undergraduates had higher intrinsic interest in English learning.

*The total motivational value and intrinsic interest motivation of female students were significantly higher than those of male students.*

Many studies have confirmed that female university students' English learning motivation is significantly higher than that of male students. This study verified the research results of Yang and Quadir [40]. Oga-Baldwin and Nakata [41] believed that effort was a mediating variable of motivation, affecting the academic performance of English learners. This study was in line with the findings of You and Dörnyei [34], confirming that girls had stronger visions for self-development than boys, and as a result, female students studied

harder than male students. Therefore, it could be seen that female students in ethnic universities made more efforts and studied harder than male students in English learning.

Kusurkar et al. [72] indicated that female minorities had stronger motivation than male minorities, especially intrinsic motivation. Not surprisingly, we found the same relationship in minority college students. Contrary to the general research, it was found that in some ethnic groups, male motivation could be higher than female motivation [39]. However, this result should consider the influence of the historical background, political status, and cultural traditions.

*Research Question 3: Motivational Impact on English Achievement*
*Intrinsic interest has a significant impact on English achievement.*

This study verified the idea that motivation promoted academic achievement. The more motivated you are, the better your academic performance [5,43]. Intrinsic interest motivation had significant predictive effects on the English academic performance of ethnic undergraduates. This study validated the findings of Almulla and Alamri [44], Taylor, Jungert, Mageau, Schattke, Dedic, Rosenfield, and Koestner [45], and Ryan and Deci [31] that intrinsic interest had a significant positive impact on academic achievement. A recent study indicated that intrinsic motivation could positively predict course satisfaction [73]. It supported the idea that intrinsic interest plays a positive role in the improvement of English learning performance. Ethnic minority students learned English from an interest in the English language itself and the culture of English-speaking countries, as well as the sustainable development of their future learning. Based on the present study, the results indicated that the intrinsic interest motivation of Han students was significantly higher than that of minority students. This was disadvantageous to minority students' learning. Therefore, parents and teachers should focus on cultivating ethnic minority students' inner interests, to encourage minority students to develop their ideal L2 self and sustainable motivation.

*Learning situation motivation has a significantly negative impact on English achievement.*

Learning situation motivation reflected the L2 learning experience based on the L2 motivational self system from the framework of Dörnyei and Ushioda [19]. The situational motivation was mainly reflected in teachers and courses [34]. The finding had some resonance with the finding of Darvin and Norton [74], which showed that students' identity and learning context could lead to negative learning results. Mercer and Dörnyei [75] believed that a teacher-friendly environment could motivate student's active engagement in learning. According to the research results, the learning situation of ethnic minority students was unfavorable to their English academic achievement. The learning situation was considered to be the strongest factor affecting the academic performance of ethnic minorities [10], and this disadvantage deeply inhibited the personal development of ethnic minority students. This kind of negative motivation influence was not only disadvantageous to educational fairness but also disadvantageous to the sustainable development of their entire education. Therefore, schools should establish favorable learning situations for minority students, and teachers should reduce the negative effects of situational factors on minority students. It is suggested that to create a circumstance in which minority students can feel that they have a place in the learning environment. This could include support groups for students from different ethnic backgrounds to share experiences and support each other. During the class, discussion on ethnic and cultural differences in learning environments could be helpful to minority students [10].

## 7. Conclusions and Implications

To conclude, this study has explored four different types of English learning motivation in Chinese ethnic universities and has particularly examined the differences in undergraduates' views according to the two background factors of ethnicity and gender. The impact

of motivation on English achievement has been investigated. Intrinsic interest motivation has a significantly positive impact on English achievement, whereas learning situation motivation has a significantly negative impact on English achievement. The particularities of minority undergraduates have been identified and systematically analyzed.

Three contributions can be summarized. First of all, in terms of topic selection, ethnic undergraduates are selected as the research objects. Compared with non-ethnic university students, the samples have particularities. This provides a reference for ethnic second language teaching and ethnic higher education.

The second contribution is theoretical innovation. In the past, researchers have focused on the perspective of social psychology and discussed the ideal L2 self too much, including its integration and instrumental motivation. In this study, the theory is deeply explored, focusing on the ideal L2 self, the ought-to L2 self, and the L2 learning experience under the L2 motivational self system [19].

The final contribution is method innovation. Through empirical research, this study adds a deep discussion of internal structure and innovates the research method.

We recommend improving the learning situation of language teaching in Chinese ethnic universities and developing cultural awareness and competence for ethnic undergraduates. Busse and Walter [50] suggested that the motivation for second language learning was largely influenced by the learning situation and was closely related to second language teachers and courses. Improvement of the language learning situation in ethnic universities could promote students' English levels and boost their English learning motivation. Improvement of the English learning situation in ethnic universities could also promote the English teaching quality. If ethnic minorities could find a sense of belonging in the present context, their motivations could be further developed [27,76]. Teachers should improve students' confidence and expectations by taking into account the cultural background of ethnic minority students and clarifying expectations, for example, through the use of ethnic role models. A multicultural exchange platform could also help minority students become close to residents, and enable native residents to have a better understanding of minority cultures [77]. Therefore, these recommendations could help improve the learning quality and sustainable development of ethnic minorities.

Interest encouragement could also be helpful for minority students to actively engage in second language learning. Second language learning should not remain a superficial interest or be judged by exam results alone. It also needs constant dedication and hard work all day long to achieve its due results and gains. According to Astin [78], although the environment plays a crucial role in the development of university students, the ultimate achievement of university students mainly depends on their involvement in the environment. The improvement of English learning motivation mainly depends on the efforts and dedication of students. At the same time, more attention should be given to minority students in minimizing their disadvantages and differences brought by family backgrounds. Schools and teachers should respond to the demands of minority students and seek to promote the fairness of education opportunities, to achieve education for all and sustainable development.

The COVID-19 pandemic has shifted students' learning from classroom to online learning, which has completely changed the current educational pattern and has had a profound impact on students' motivation to learn [79]. Studying alone at home online has brought great challenges to learning motivation [80]. Student's learning motivation and enthusiasm have decreased [81]. In addition, students' achievement goals, engagement, and perception of success have declined significantly [82]. In this case, teachers and parents should strive to reduce the negative impact of COVID-19 restrictions on students and cultivate students' learning autonomy and enthusiasm [81]. Given the change in learning pattern, flipped classrooms could be adopted to improve students' learning autonomy, to improve students' learning [79]. Reducing negative emotions and increasing positive emotions could also be helpful to help with motivations [83].

The previous studies on motivation were mostly conducted in multi-ethnic countries with European or American cultures, and most of the studies involved English-speaking countries, such as the United States of America and Australia. However, there were few studies on motivation conducted in non-English-speaking countries [84]. This study investigates the second language learning motivation of ethnic minority students in a non-English-speaking country, which provides an important reference for future researchers in the world.

## 8. Limitations and Future Work

This study has made a comparison of the English learning motivation of Han and minority undergraduates from ethnic universities. However, a comparison among different ethnic groups is not involved or discussed. In the future, the English learning motivation of university students of different ethnic groups could be classified and compared. Family background and educational experience could be used as reference variables to further explore second language learning motivation. Follow-up qualitative research such as interviews could be carried out to explore students' living experiences and especially their views on reasons for learning English as a second language.

**Author Contributions:** Conceptualization, S.J. and J.W.; methodology, S.J.; software, X.M.; validation, J.W., Z.Y. and D.J.; formal analysis, S.J.; investigation, X.M.; resources, S.J.; data curation, J.W.; writing—original draft preparation, S.J.; writing—review and editing, J.W., Z.Y. and D.J.; visualization, X.M.; supervision, J.W.; project administration, S.J.; funding acquisition, S.J. All authors have read and agreed to the published version of the manuscript.

**Funding:** This research was funded by the National Social Science Fund of China (Grant No. BMA200041) and the Beijing Society of Education (Grant No. CYYB2021-579).

**Institutional Review Board Statement:** The study was conducted according to the guidelines of the Declaration of Helsinki, and approved by the College of Education, Minzu University of China.

**Informed Consent Statement:** Informed consent was obtained from all subjects involved in the study.

**Data Availability Statement:** The data presented in this study are available on request from the author. The data are not publicly available due to ethical considerations.

**Conflicts of Interest:** The authors declare no conflict of interest.

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
