# Peer review of "Motivation and Its Impact on Language Achievement: Sustainable Development of Ethnic Minority Students’ Second Language Learning"

_sustainability, doi:10.3390/su14137898_

Round 1
Reviewer 1 Report
The reviewed paper addresses a relevant issue of ethnic minority undergraduate students' motivation for English learning. It can contribute to the discussion of various factors affecting motivation for language learning. However, there are several isssues which are to be specified or revised.
1) the description of motivations types and motivation structure has to be significantly systematized. In the literature review several motivation types are described, but the criteria for their selecting and structuring are not quite clear.
2) in the literature review
2) participant groups are to be described in more detail, particularly the ethnic minority group. Does it include the students of the same or different ethnic origins? If the students belonged to different ethnic groups, how correct it is to join them into one sample?
3) the study analyzes four types of motivation. It is not quite clear how these types were determined.
Author Response
Cover Letter 1
Author's Reply to the Review Report (Reviewer 1)
Dear Madam/ Sir,
Thank you very much for reviewing the manuscript and your valuable time. Your suggestions are very valuable and helpful for us to improve the quality of the manuscript. The point to point responses are as follows, you can also find it on the attachment and the revised version of the manuscript in the “Track Changes” Word.
1) The description of motivations types and motivation structure has to be significantly systematized. In the literature review several motivation types are described, but the criteria for their selecting and structuring are not quite clear.
Reply: Please see the revised version of 2.1. Motivation Composition (See Line 104-168 on the revised version)
Early Research on language learning motivation follows Gardner's (1985) social educational model, which divides second language learning motivation into integrativeness and instrumentality [1-3]. Some scholars also take internal motivation and external motivation as a paradigm to discuss language learning based on Self-determination theory[4-6]. However, with the development of globalization, English has broken away from the limitations of traditional regions. It seems difficult to solve the fundamental problem of English learning motivation only from the perspective of integration and instrumentality, and even intrinsic and extrinsic motivation [7]. Dörnyei and Ushioda [8] redefined language learning motivation from the perspective of self-identity and proposed a “L2 Motivational Self System”, which includes three aspects, the Ideal L2 Self, the Ought-to L2 Self and the L2 Learning Experience [9].
Gardner and Lambert [10] suggested that instrumental and integrative motivation were the main motivation of second language learning. The instrumental motivation is defined as the utilitarian acquisitions of second language proficiency, such as getting a higher salary or a better job. While integrative motivation is defined as the willingness to be like valued members of the language community [11]. Since the orientation was proposed, instrumentality and integrativeness have been the focus of the research on second language learning motivation. In recent years, studies on second language learning motivation have shifted their focus from instrumental and integrative discussions to more complex socio-cultural situations and interactions between learners. The influence of learning situation and personal vision on second language learning motivation has become more and more important.
Csizér and Dörnyei [12] stated that integrativeness is the most important motivation for second language learning, mediating the influence of all the other factors. Aoki [13] found that Chinese students held higher integrative motivation than that of other countries. A recent study [14] asserted that most second language learners possessed integrative learning motivation, and people with high integrative motivation could get high achievement easily. However, an opposite voice indicated that instrumental learning motivation were the preferences for second language learners [15]. This argument was supported by Long, et al. [16] that examination was the main cause of the change of students' English learning motivation. Dos Santos [17] advocated that both integrative motivation and instrumental motivation could promote learning. This idea echoed the finding of Gardner and MacIntyre [18] that motivation had dual components, integrativeness and instrumentality, both of them could beef up language learning.
Integrative motivation and instrumental motivation are not mutually exclusive, generally, second language learning involves a composition of various levels. Therefore, it is hard to classify the language learning motivation to a precise side, the importance of which relies on the situation of the study [19]. At the same time, intrinsic motivation and extrinsic motivation have become another perspectives of language learning research [20, 21]. It is undeniable that the research on the internal and external division of motivation has important implications for individuals’ willpower and academic achievement [22, 23]. However, this division makes it difficult for us to realize the learning experience of second language learners, such as learning situations, which is particularly important for minority students [24]. In fact, both intrinsic and extrinsic motivation could find their explanations from the three levels of “L2 Motivational Self System” [8]. Therefore, using the theory of “L2 Motivational Self System” can discover the unsolved mysteries of language learning motivation in the past.
In the past decade, “L2 Motivational Self System” has been widely applied in the global environment, and has become the mainstream paradigm for the study of language motivation [20]. You and Dörnyei [25] conducted a large-scale survey on Chinese language learners from the perspective of the “L2 Motivational Self System”, and found that the motivation types of Chinese second language learners are instrumentality-promotion, cultural interest, travelling, instrumentality-prevention, parental expectation and expected efforts. Moskovsky, et al. [26] explored the relationship between Dörnyei (2009)’ L2 Motivational Self System and the second language proficiency of Saudi L2 learners. The study classified second language learning motivation according to the level of the Ideal L2 Self, the Ought-to L2 Self and the L2 Learning Experience [8, 9], indicating that the components of L2 Motivational Self System are good predictors of learners’ intention to learn. Gao, et al. [27] found seven main types of English learning motivation for university students in China: intrinsic interest, immediate achievement, going abroad, learning context, social responsibility, personal development and information media, which were in line with the aspects of “L2 Motivational Self System”.
Why it is called motivation structure? (See Line 246-258 on the revised version)
The previous study on motivation composition is usually called motivation type, which is a classification based on exploratory factor analysis (EFA)[28, 29]. EFA aims to confirm the factor structure of the scale, which is biased towards theoretical output, but lacks the test of theoretical framework[30]. Confirmatory factor analysis (CFA) is usually used to verify the appropriateness of the measurement model. CFA is a basic component of structural equation modeling (SEM)[31]. CFA analysis is often tested by structural equation modeling, which can test the authenticity and appropriateness of construct validity. [32] On the basis of EFA, CFA is added to the present study in order to obtain the best factor structure of the questionnaire and establish the construct validity of the questionnaire. CFA can be used to test the structural validity of motivation, so as to determine the classification structure of motivation more accurately[33]. Therefore, the combination of CFA and EFA goes beyond the exploration of motivation types[34]. Here, we call it the internal structure of motivation. 2) In the literature review |participant groups are to be described in more detail, particularly the ethnic minority group. Does it include the students of the same or different ethnic origins? If the students belonged to different ethnic groups, how correct it is to join them into one sample?
Reply: (See Line 331-341 on the revised version)
The participants of ethnic minorities selected in the study come from different ethnic groups, including Chinese Mongolian, Tibetan and other ethnic groups, mainly representing the macro group of ethnic minorities. This study does not compare each ethnic group separately. Ethnic minority college students often come from frontier ethnic areas, usually located in remote and economically underdeveloped areas, which is far away from the developed eastern coastal areas [35]. It is generally believed that the majority students have obvious advantages over minority students in second language learning [36]. The research on ethnic education takes the whole ethnic minorities as a sample to compare with the mainstream ethnic Han [37] . Therefore, we can take minority college students as a research subject in this study. The minority students are the combinations of 55 ethnic groups compared to Han college students.
3) The study analyzes four types of motivation. It is not quite clear how these types were determined.
Reply: (See Line 520-525 on the revised version)
According to the questionnaire structure of Gao, Zhao, Cheng and Zhou [27], the study firstly constructed the initial items of the questionnaire on the English learning motivation of ethnic undergraduates. Secondly, four dimensions of motivation were constructed through exploratory factor analysis (EFA). In addition, combined with confirmatory factor analysis (CFA), this paper tested its structural validity, and finally determined the four structures of ethnic undergraduates’ English learning motivation.
References
- Tremblay, P. F.; Gardner, R. C., Expanding the motivation construct in language learning. The modern language journal 1995, 79, (4), 505-518.
- Gardner, R. C.; MacIntyre, P. D., A student's contributions to second-language learning. Part II: Affective variables. Language teaching 1993, 26, (1), 1-11.
- MacIntyre, P. D.; MacKinnon, S. P.; Clément, R., Toward the development of a scale to assess possible selves as a source of language learning motivation. Motivation, language identity and the L2 self. Bristol: Multilingual Matters 2009, 193-214.
- Noels, K. A., New orientations in language learning motivation: Towards a model of intrinsic, extrinsic, and integrative orientations and motivation. Motivation and second language acquisition 2001, 23, 43-68.
- Ryan, R. M.; Deci, E. L., Overview of self-determination theory: An organismic dialectical perspective. Handbook of self-determination research 2002, 2, 3-33.
- Deci, E. L.; Koestner, R.; Ryan, R. M., Extrinsic rewards and intrinsic motivation in education: Reconsidered once again. Review of educational research 2001, 71, (1), 1-27.
- Dörnyei, Z.; Ryan, S., The psychology of the language learner revisited. Routledge: 2015.
- Dörnyei, Z.; Ushioda, E., Motivation, language identity and the L2 self. Multilingual Matters: Bristol, Blue Ridge Summit, 2009; Vol. 36.
- Dörnyei, Z., The psychology of the language learner: Individual differences in second language acquisition. New Jersey: Mahwah 2005.
- Gardner, R. C.; Lambert, W. E., Attitudes and Motivation in Second-Language Learning. 1972.
- Gardner, R. C.; Lambert, W. E., Motivational variables in second-language acquisition. Canadian Journal of Psychology 1960, 13, 266-272.
- Csizér, K.; Dörnyei, Z., The internal structure of language learning motivation and its relationship with language choice and learning effort. The modern language journal 2005, 89, (1), 19-36.
- Aoki, N., The challenge of motivation: Teaching Japanese kanji characters to students from diverse language backgroundsl. Bridging Transcultural Divides: Asian Languages and Cultures in Global Highter Education 2012, 131.
- Samad, A. A.; Etemadzadeh, A.; Far, H. R., Motivation and language proficiency: Instrumental and integrative aspects. Procedia-Social and Behavioral Sciences 2012, 66, 432-440.
- Suryasa, W.; Prayoga, I.; Werdistira, I., An analysis of students motivation toward English learning as second language among students in Pritchard English academy (PEACE). International journal of social sciences and humanities 2017, 1, (2), 43-50.
- Long, C.; Ming, Z.; Chen, L., The Study of Student Motivation on English Learning in Junior Middle School--A Case Study of No. 5 Middle School in Gejiu. English language teaching 2013, 6, (9), 136-145.
- Dos Santos, L. M., The Relationship between Social Identity and Foreign Language Learning Motivation: The Sustainability of Heritage Language Learners. Sustainability 2021, 13, (23), 13102.
- Gardner, R. C.; MacIntyre, P. D., An instrumental motivation in language study: Who says it isn't effective? Studies in second language acquisition 1991, 13, (1), 57-72.
- Gardner, R. C.; Lambert, W. E., Attitudes and motivation in second-language learning. Rowley, MA: Newbury House: 1972.
- Boo, Z.; Dörnyei, Z.; Ryan, S., L2 motivation research 2005–2014: Understanding a publication surge and a changing landscape. System 2015, 55, 145-157.
- Shaikholeslami, R.; Khayyer, M., Intrinsic motivation, extrinsic motivation, and learning English as a foreign language. Psychological reports 2006, 99, (3), 813-818.
- Ryan, R. M.; Deci, E. L., Intrinsic and extrinsic motivation from a self-determination theory perspective: Definitions, theory, practices, and future directions. Contemporary educational psychology 2020, 61, 101860.
- Ng, C. F.; Ng, P. K., A review of intrinsic and extrinsic motivations of ESL learners. International Journal of Languages, Literature and Linguistics 2015, 1, (2), 98-105.
- Próspero, M.; Russell, A. C.; Vohra-Gupta, S., Effects of Motivation on Educational Attainment: Ethnic and Developmental Differences Among First-Generation Students. Journal of Hispanic Higher Education 2012, 11, (1), 100-119.
- You, C. J.; Dörnyei, Z., Language learning motivation in China: Results of a large-scale stratified survey. Applied Linguistics 2016, 37, (4), 495-519.
- Moskovsky, C.; Assulaimani, T.; Racheva, S.; Harkins, J., The L2 motivational self system and L2 achievement: A study of Saudi EFL learners. The Modern Language Journal 2016, 100, (3), 641-654.
- Gao, Y.; Zhao, Y.; Cheng, Y.; Zhou, Y., Relationship Between English Learning Motivation Types and Self-Identity Changes Among Chinese Students. TESOL Quarterly 2007, 41, (1), 133-155.
- Liu, H.; Gao, L.; Fang, F., Exploring and Sustaining Language Teacher Motivation for Being a Visiting Scholar in Higher Education: An Empirical Study in the Chinese Context. Sustainability 2020, 12, (15), 6040.
- Han, F., The Relations Between Motivation, Strategy Use, Frequency, and Proficiency in Foreign Language Reading: An Investigation With University English Language Learners in China. SAGE Open 2021, 11, (2), 21582440211008423.
- Bandalos, D. L.; Finney, S. J., Factor analysis: Exploratory and confirmatory. In The reviewer’s guide to quantitative methods in the social sciences, Routledge: 2018; pp 98-122.
- Brown, T. A., Confirmatory factor analysis for applied research. Guilford publications: 2015.
- Ullman, J. B.; Bentler, P. M., Structural equation modeling. Handbook of Psychology, Second Edition 2012, 2.
- Martin, A. J., Examining a multidimensional model of student motivation and engagement using a construct validation approach. British Journal of Educational Psychology 2007, 77, (2), 413-440.
- Wang, C.; Bai, B., Validating the instruments to measure ESL/EFL learners' self-efficacy beliefs and self-regulated learning strategies. TESOL quarterly 2017, 51, (4), 931-947.
- Xiong, W., Ethnic minority-serving institutions: Higher education case studies from the United States and China. Springer: 2020.
- Adamson, B.; Xia, B., A case study of the College English Test and ethnic minority university students in China: negotiating the final hurdle. Multilingual Education 2011, 1, 1-11.
- Sude; Yuan, M.; Dervin, F., Studying Chinese Minority Education. In An Introduction to Ethnic Minority Education in China: Policies and Practices, Springer Berlin Heidelberg: Berlin, Heidelberg, 2020; pp 1-8.

Reviewer 2 Report
Good piece of research. The sample is good enough and it covers a minority that is representative for the authors and their audience.
A distinction between L2 and FL (or additional languages) should be done, clarifying if these students have English as a FL or a 2L (the answer is clear but it would be more if said by the authors). Both FL and 2L are used.
There is no reference to COVID and if this has had any influence on the levels of motivation of these students. I’m quite positive it has
A minor revision of spelling (See University in Table 1)
The authors should mention how this study can be applied in other contexts clearly, not only in China (there are 56 ethnic groups and the possibility of getting common results in others could be interesting) but if the same results could happen in other contexts as a kind of hypothesis for future researches.
It would also be interesting if the authors described if in the gender section it could be different in other ethnics due to political reasons, traditions or any other aspects that could affect.
Regarding the participation (800 questionnaires distributed and 776 returned), do please clarify if the participation was voluntary or if, on the hand, participants were, in any sense, obliged to participate.
Author Response
Cover Letter 2
Author's Reply to the Review Report (Reviewer 2)
Dear Madam/ Sir,
Thank you very much for reviewing the manuscript and thank you for your valuable time. Your suggestions are very valuable and helpful for us to improve the quality of the manuscript. The point-to-point responses are as follows, you can also find it on the attachment and the revised version of the manuscript in the “Track Changes” Word.
1) A distinction between L2 and FL (or additional languages) should be done, clarifying if these students have English as a FL or a 2L (the answer is clear but it would be more if said by the authors). Both FL and 2L are used.
Reply: Thank you for your kind reminding, we agree with your advice and have changed the term “FL” into “L2“ through the whole article. (See the revised version)
2) There is no reference to COVID and if this has had any influence on the levels of motivation of these students. I’m quite positive it has.
Reply: We agree with your suggestions that COVID-19 has positive influence on the levels of motivation of these students. In accordance with your implications, we conduct the following paragraph.
The COVID-19 pandemic has shifted students' learning from classroom to online learning, which has completely changed the current educational pattern and has a profound impact on students' motivation to learn [1]. Studying alone at home online has brought great challenges to learning motivation [2]. Students' learning motivation and enthusiasm have decreased [3]. In addition, students' achievement goals, engagement and perception of success have declined significantly [4]. In this case, teachers and parents should strive to reduce the negative impact of Covid-19 restrictions on students and cultivate students' learning autonomy and enthusiasm [3]. In view of the change of new learning pattern, flipped classroom could be adopted to improve students' learning autonomy, so as to improve students' learning effect [1]. Reducing negative emotions and increasing positive emotions could also be helpful to cope with the motivations [5]. (See Line 696-706 on the revised version)
3) A minor revision of spelling (See University in Table 1)
Reply: We have checked the spelling of “University” in table 1. (See the revised version)
4) The authors should mention how this study can be applied in other contexts clearly, not only in China (there are 56 ethnic groups and the possibility of getting common results in others could be interesting) but if the same results could happen in other contexts as a kind of hypothesis for future researches.
Reply: The application in other contexts is very helpful for our further explanation of our study.
The previous studies about motivations were mostly conducted in multi-ethnic countries with European or American cultures, and most of the studies involved English-speaking countries, such as the United States of America and Australia. However, there were few studies on motivation conducted in non-English speaking countries [6]. This study investigates the second language learning motivation of ethnic minority students in a non-English speaking country, which provides an important reference for future researchers in the world. (See the revised version: Line 707-713)
5) It would also be interesting if the authors described if in the gender section it could be different in other ethnics due to political reasons, traditions or any other aspects that could affect.
Reply:
Kusurkar, et al. [7] indicated that female minorities had stronger motivation than male minorities, especially intrinsic motivation. Not surprisingly, we found the same relationship in minority college students. Contrary to the general research, it was found that in some ethnic groups, male motivation could be higher than female motivation [8]. However, this result should consider the influence of the historical background, political status and cultural traditions. (See the revised version: Line 608-613)
6) Regarding the participation (800 questionnaires distributed and 776 returned), do please clarify if the participation was voluntary or if, on the hand, participants were, in any sense, obliged to participate.
Reply: Thank you for reminding and the clarification of this point is fairly important.
This study was anonymous and the participation was voluntary. (See the revised version: Line 327-328)
References
- Campillo-Ferrer, J. M.; Miralles-Martínez, P., Effectiveness of the flipped classroom model on students’ self-reported motivation and learning during the COVID-19 pandemic. Humanities and Social Sciences Communications 2021, 8, (1), 1-9.
- Rahm, A.-K.; Töllner, M.; Hubert, M. O.; Klein, K.; Wehling, C.; Sauer, T.; Hennemann, H. M.; Hein, S.; Kender, Z.; Günther, J., Effects of realistic e-learning cases on students’ learning motivation during COVID-19. PloS one 2021, 16, (4), e0249425.
- Zaccoletti, S.; Camacho, A.; Correia, N.; Aguiar, C.; Mason, L.; Alves, R. A.; Daniel, J. R., Parents’ Perceptions of Student Academic Motivation During the COVID-19 Lockdown: A Cross-Country Comparison. Frontiers in Psychology 2020, 11.
- Daniels, L. M.; Goegan, L. D.; Parker, P. C., The impact of COVID-19 triggered changes to instruction and assessment on university students’ self-reported motivation, engagement and perceptions. Social Psychology of Education 2021, 24, (1), 299-318.
- Wang, K.; Goldenberg, A.; Dorison, C. A.; Miller, J. K.; Uusberg, A.; Lerner, J. S.; Gross, J. J.; Agesin, B. B.; Bernardo, M.; Campos, O., A multi-country test of brief reappraisal interventions on emotions during the COVID-19 pandemic. Nature Human Behaviour 2021, 5, (8), 1089-1110.
- Isik, U.; Tahir, O. E.; Meeter, M.; Heymans, M. W.; Jansma, E. P.; Croiset, G.; Kusurkar, R. A., Factors Influencing Academic Motivation of Ethnic Minority Students: A Review. SAGE Open 2018, 8, (2), 2158244018785412.
- Kusurkar, R.; Ten Cate, T. J.; Vos, C.; Westers, P.; Croiset, G., How motivation affects academic performance: a structural equation modelling analysis. Advances in health sciences education 2013, 18, (1), 57-69.
- D’Lima, G. M.; Winsler, A.; Kitsantas, A., Ethnic and Gender Differences in First-Year College Students’ Goal Orientation, Self-Efficacy, and Extrinsic and Intrinsic Motivation. The Journal of Educational Research 2014, 107, (5), 341-356.

Reviewer 3 Report
The article deals with the topical issue of motivation and its importance for the achievements of ethnic minority learners in China in studying English as a foreign language. The findings indicate four types of motivation related to intrinsic interest, the learning situation, personal development, and international communication. The research work is informative and interesting to read and I believe it would be beneficial to a large audience of readers.
The introduction presents background information related to the necessity to study the English learning motivation of minority college students in China, which is a multi-ethnic country with as many as 56 ethnic groups.
The Literature review gives an extensive overview of a number of relevant contemporary studies and the overall trend of shifting from simple individual psychological factors to complex socio-cultural situations and interactions between individuals is emphasized.
The authors pose the following three questions as regards their aim to explore the motivation of ethnic minority university students in English learning, focusing on the internal structure, ethnic and gender differences, and the motivational impact on the learners’ English achievement:
· What are the internal structures of English learning motivation for undergraduates in Chinese ethnic universities?
· Are there any differences between dominant ethnic groups (Han) and ethnic minority students, male students, and female students?
What is the motivational impact on minority students’ English academic performance?
The Theoretical frameworks are clear and factual, and the Materials and methods section describes the participants, instruments, and the procedure.
The Results section reports the analyses made comprehensively and explicitly, namely an item analysis, an exploratory factor analysis of the internal structure, a confirmatory factor analysis, and a reliability analysis. Also, the internal relationship among the factors of English learning motivation is explored and students' motivation to learn English is characterized, involving a description of English learning motivation in each dimension, ethnic and gender differences in motivation, and motivational influence on learners’ achievement in English.
In the Discussions section, the authors provide answers to the questions they initially asked:
Q1. Four motivation components were found regarding ethnic university undergraduates.
Q2. The intrinsic interest motivation of Han students was higher than that of minority students and the total motivational value and intrinsic interest motivation of female students were significantly higher than that of male students.
Q3. Intrinsic interest has an important impact on English achievement while learning situation motivation has a significantly negative impact on it.
The Conclusions section summarizes the authors’ three major contributions made in the study, and the Future work suggests ideas for further research involving the classification and comparison of different ethnic groups regarding university students’ second language learning motivation based on their family background and educational experience.
The authors have included a sufficient number of up-to-date references in their research.
The language of the article is excellent. There are just a few insignificant things I noticed, for example on line 95: execute the original wished and desires (wishes) or on line 127: to an precise side (a), etc.
In conclusion, I would recommend that the article be published in its present form.
Author Response
Cover Letter 3
Author's Reply to the Review Report (Reviewer 3)
Dear Madam/ Sir,
Thank you very much for reviewing the manuscript and thank you for your valuable time. The point-to-point responses are as follows, you can also find it on the attachment and the revised version of the manuscript in the “Track Changes” Word.
Reviewer’s advice: The language of the article is excellent. There are just a few insignificant things I noticed, for example on line 95: execute the original wished and desires (wishes) or on line 127: to an precise side (a), etc. In conclusion, I would recommend that the article be published in its present form.
Reply: Thank you for your affirmation and recommendation, we agree with your advice and have checked the word “wishes” and “to a precise side “. Moreover, we have checked the spelling of the words through the whole article. (See the revised version)
All in all, thank you so much for you time and hard work. Wish you a pleasant day!

Round 2
Reviewer 1 Report
Dear authors,
thank you very much for your commentaries and corrections.
Author Response
Cover Letter 1
Author's Reply to the Review Report (Reviewer 1-2nd round)
Dear Madam/ Sir,
Thank you very much for your commentaries and help. We would like to express our gratitude for your review work and corrections.
1) English language and style are fine/minor spell check required.
Reply: We have checked the spelling of the words and have adjusted some of the expressions in this article, such as the abstract. For the rest of the checking, please see the revised version.
Example-Abstract: This study examined the English learning motivation of Chinese ethnic university students. A sample of 776 undergraduates from three representative ethnic universities participated in this research. The findings indicated four types of English learning motivation concerning intrinsic interest, learning situation, personal development, and international communication. There were statistically significant differences between ethnic minority and Han students, and between male and female students. Moreover, the intrinsic interest motivation of Han students was significantly higher than that of ethnic minority students, and female students' overall motivation and personal development motivation were significantly higher than that of male students. Intrinsic interest motivation had a significantly positive impact on English achievement, whereas learning situation motivation had a significantly negative impact. These findings highlight the improvement of the learning situation and encouragement of intrinsic interest to enhance minority students’ second language learning and sustainable development. Further research on English as a second language should consider the influence of family and ethnic background.
